# Whole-Genome Analysis of Influenza A(H3N2) and B/Victoria Viruses Detected in Myanmar during the COVID-19 Pandemic in 2021

**DOI:** 10.3390/v15020583

**Published:** 2023-02-20

**Authors:** Irina Chon, Reiko Saito, Yadanar Kyaw, Moe Myat Aye, Swe Setk, Wint Wint Phyu, Keita Wagatsuma, Jiaming Li, Yuyang Sun, Teruhime Otoguro, Su Mon Kyaw Win, Sayaka Yoshioka, Nay Chi Win, Lasham Di Ja, Htay Htay Tin, Hisami Watanabe

**Affiliations:** 1Division of International Health, Graduate School of Medical and Dental Sciences, Niigata University, Niigata 951-8510, Japan; 2Infectious Diseases Research Center of Niigata University (IDRC), Graduate School of Medical and Dental Sciences, Niigata University, Niigata 951-8510, Japan; 3Respiratory Medicine Department, Thingangyun General Hospital, Yangon 110-71, Myanmar; 4National Health Laboratory, Department of Medical Services, Dagon Township, Yangon 111-91, Myanmar; 5Japan Society for the Promotion of Science, Tokyo 102-0083, Japan; 6Infectious Diseases Research Center of Niigata University in Myanmar (IDRC), Yangon 111-91, Myanmar

**Keywords:** influenza, A(H3N2), B/Victoria, Myanmar, whole-genome analysis, COVID-19, SARS-CoV-2

## Abstract

An influenza circulation was observed in Myanmar between October and November in 2021. Patients with symptoms of influenza-like illness were screened using rapid diagnostic test (RDT) kits, and 147/414 (35.5%) upper respiratory tract specimens presented positive results. All RDT-positive samples were screened by a commercial multiplex real-time polymerase chain reaction (RT-PCR) assay, and 30 samples positive for influenza A(H3N2) or B underwent further typing/subtyping for cycle threshold (C*t*) value determination based on cycling probe RT-PCR. The majority of subtyped samples (n = 13) were influenza A(H3N2), while only three were B/Victoria. Clinical samples with low C*t* values obtained by RT-PCR were used for whole-genome sequencing via next-generation sequencing technology. All collected viruses were distinct from the Southern Hemisphere vaccine strains of the corresponding season but matched with vaccines of the following season. Influenza A(H3N2) strains from Myanmar belonged to clade 2a.3 and shared the highest genetic proximity with Bahraini strains. B/Victoria viruses belonged to clade V1A.3a.2 and were genetically similar to Bangladeshi strains. This study highlights the importance of performing influenza virus surveillance with genetic characterization of the influenza virus in Myanmar, to contribute to global influenza surveillance during the COVID-19 pandemic.

## 1. Introduction

Influenza is a highly transmissible disease that poses a serious threat to public health and is known to cause mortality in children under the age of 5 years [1]. Monitoring the spread of influenza viruses is essential for the annual planning of influenza virus prevention and control in each country. At the end of 2019, severe acute respiratory syndrome coronavirus 2 (SARS-CoV-2) emerged, rapidly spreading worldwide and causing the COVID-19 pandemic [2]. Since the start of the COVID-19 pandemic, influenza surveillance has undergone radical changes, with extremely low case detection rates in 2020 and 2021 worldwide [3], which are probably associated with non-pharmaceutical measures, such as travel restrictions, wearing masks, using sanitizer and social distancing, implemented by each government [4].

In 2015, with the support of the Japan Agency for Medical Research and Development (AMED), several collaborative centers with Niigata University were established in Myanmar. From 2015 to 2019, over 2000 respiratory samples were collected at the surveillance sites for further testing and molecular-epidemiological analysis. Through this project, we clarified influenza seasonality in Myanmar, characterized influenza viruses circulating in the country, identified unique influenza strains in Myanmar and proved international influenza dissemination links with many countries [5,6,7,8,9,10]. These findings highlight the importance of investigating the spread of influenza in Myanmar.

In Myanmar, the first two COVID-19-positive cases were confirmed on 23 March 2020, and subsequently, two community outbreaks occurred [11]. The third huge wave of COVID-19 in Myanmar that was caused by delta variants was observed in the rainy season of 2021, which started in June and gradually diminished until January 2022 [12]. The peak of the third wave was observed on 13 July 2021, with 7083 cases, and a maximum of 397 fatal cases were detected on 1 August 2021 [12]. However, according to the country report to the Influenza Laboratory Surveillance Information (Flu-Net) run by the World Health Organization (WHO) Global Influenza Surveillance and Response System, influenza activity in Myanmar was low in 2020–2021, and a very limited number of cases were detected in October 2020 and October 2021 [3]. The reduced circulation of influenza viruses after the COVID-19 pandemic could affect the magnitude of the subsequent influenza infection because of the prolonged absence of natural exposure to influenza viruses [13].

Here, we reported the epidemiology and genetic characteristics of influenza A(H3N2) and B/Victoria viruses that circulated in Myanmar during October and November 2021, after the large community outbreaks of the delta variant of SARS-CoV-2. The WHO recommends Southern Hemisphere (SH) formulations of vaccines for Tropical Asia; therefore, we analyzed the matches with SH vaccines in the same year using whole-genome sequencing and assessed the genetic relations with the influenza strains circulating in other countries during the same period [14].

## 2. Materials and Methods

### 2.1. Sample Collection

Patients with influenza-like illness (ILI) symptoms (body temperature ≥ 37.5 °C, cough or sore throat, and rhinorrhea, malaise, chills, headache, muscle or body aches, cough, fatigue, or nausea), who visited outpatient clinics in Yangon Region between October and December 2021, were eligible to participate in this study. Informed consent was obtained from the patients, and clinical information such as name, age, sex, address, date of symptom onset, date of clinic visit, anti-influenza vaccination history, antiviral drug medication history, and symptoms were recorded. Patients were screened using a rapid diagnostic test kit (RDT) (Quick Navi-Flu+RSV, Denka Seiken Co. Ltd., Tokyo, Japan), and nasal or throat swabs were collected from RDT-positive influenza A, B, or respiratory syncytial virus (RSV) patients. Swabs were placed in a viral transport medium (VTM) and kept at −20 °C. Samples were shipped to the Division of International Health Graduate School of Medical and Dental Sciences of Niigata University, Japan, where they were stored at −80 °C. Data of COVID-19 activity in Myanmar were retrieved from the Myanmar COVID-19 surveillance dashboard of the Ministry of Health in Myanmar, which is updated daily and publicly available [12].

### 2.2. Laboratory Confirmation

#### 2.2.1. Multiplex PCR Screening

All received swabs were tested by a BioFire^®^ FilmArray^®^ respiratory panel (Salt Lake City, UT, USA). The device is fully automated with the ability to perform nucleic acid purification, amplification, multiplexed PCR, and melting analysis, and generates a report for the qualitative detection of various respiratory pathogens [15,16]. Screening was performed according to the manufacturer’s instructions [17]. An aliquot of 300 µL of the original sample swabs was uploaded into the cassette with primers for possible respiratory viral and bacterial pathogens—Adenovirus, Coronavirus 229E, Coronavirus HKU1, Coronavirus NL63, Coronavirus OC43, Severe Acute Respiratory Syndrome Coronavirus 2 (SARS-CoV-2), Human Metapneumovirus, Human Rhinovirus/Enterovirus, Influenza A, including subtypes H1, H3 and H1-2009, Influenza B, Parainfluenza Virus 1, Parainfluenza Virus 2, Parainfluenza Virus 3, Parainfluenza Virus 4, Respiratory Syncytial Virus, *Bordetella parapertussis*, *Bordetella pertussis*, *Chlamydia pneumoniae* and *Mycoplasma pneumoniae*) [17]. After approximately 45 min, the results were displayed by BioFire^®^ FilmArray^®^ Software 2.0, including that of quality control. Influenza-positive samples were subjected to an in-house-developed cycling probe real-time PCR (RT-PCR) assay using subtype/lineage-specific primers and probes.

#### 2.2.2. RNA Extraction

Total RNA was extracted from 140 µL of clinical influenza-positive samples using a QIAamp Viral RNA Mini Kit (QIAGEN, Hilden, Germany) according to the manufacturer’s instructions. Complementary DNA (cDNA) synthesis was performed using influenza A and B universal primers (Uni–11 and Uni–12) and reverse transcription reagents (Invitrogen, Carlsbad, CA, USA) as previously reported [6,18].

#### 2.2.3. Cycling Probe Real-Time PCR for Subtyping, Lineage Detection and C*t* Value Evaluation

Clinical samples, which were obtained using synthesized cDNA, were subjected to cycling probe RT-PCR that was exclusively established in our department. Two fluorescent-labeled chimeric RNA-DNA probes with 6-carboxyfluorescein (FAM) and 6-carboxy-X-rhodamine (ROX) were designed to detect single-nucleic acid polymorphisms (SNPs) in the target DNA sequences of influenza A(H1N1)pdm09, A(H3N2), B/Victoria, and B/Yamagata [19,20,21]. RT-PCR was performed with 1 µL of sample cDNA under the following conditions: for initial denaturation at 95 °C for 10 s, denaturation at 95 °C for 5 s, primer annealing at 57 °C for 10 s, and extension/detection of fluorescence at 72 °C for 20 s. Denaturation and subsequent steps were performed for up to 50 cycles for A(H3N2) and 45 cycles for influenza B. A C*t* cut-off of value less than 38 was set for both A(H3N2) and influenza B.

#### 2.2.4. Genetic Analysis by Next-Generation Sequencing

Clinical samples that showed C*t* values less than 38 by cycling probe RT-PCR were eligible for next-generation sequencing (NGS). Amplification of the whole-genome was performed using MBTuni-12-R (5′-ACGCGTGATCAGCRAAAGCAGG-3′) and MBTuni-13 (5′-ACGCGTGATCAGTAGAAACAAGG-3′) primers for influenza A and 13 primers for influenza B [22,23,24]. For both viruses, multiplex PCR was performed using the Superscript III one-step reverse transcription PCR kit with Platinum Taq (Invitrogen) [22,23]. Libraries were prepared using an enzymatic QIAseq FX DNA Library CDI Kit (QIAGEN, Hilden, Germany). The genome was enzymatically fragmented into small parts of approximately 250 bp. Platform-specific adapters were ligated to both ends of the DNA fragments. To ensure maximum yield, libraries were amplified and purified using the Agencourt AMpure XP Kit (Immunotech A Beckman, Indianapolis, IN, USA). The purified products were examined with the Agilent 4200 TapeStation (Agilent Technologies, Waldbronn, Germany) using the D1000 ScreenTape system. The concentration of each purified product was quantified with a QubitTM Flex Fluorometer (Thermo Fisher Scientific, Waltham, MA, USA) using the Qubit-iTTM 1X dsDNA HS assay (Thermo Fisher Scientific, Waltham, MA, USA). Sequences were generated using an iSeq100 machine (Illumina, San Diego, CA, USA) and genetic sequence reads were obtained as FASTQ files.

#### 2.2.5. NGS Data Analysis

Using CLC Genomics Workbench software v.20.0.2 (QIAGEN, Hilden, Germany) sequencing results were assembled and aligned based on the references A/H3N2/Louisiana/50/2017 (accession numbers CY244750-57 for 8 segments) and B/Victoria/Washington/02/2019 (accession numbers MN155748-55 for 8 segments). Generated genes were deposited in the public database Global Initiative on Sharing All Influenza Data (GISAID) (www.gisaid.org) (Appendix A). To analyze amino acid changes, whole genomes were compared with the WHO-recommended SH vaccine strains from the 2021 and 2022 seasons [25,26]. For influenza A(H3N2), the A/Hong Kong/45/2019-like virus (accession number EPI ISL 349926) was recommended for the 2021 season and the A/Darwin/6/2021-like virus (accession number EPI ISL 5332906) was recommended for the 2022 season [25,26]. For the influenza B/Victoria lineage, the WHO recommended the B/Washington/02/2019-like virus (accession number EPI ISL 347829) for the 2021 season and B/Austria/1359417/2021-like virus (accession number EPI ISL 2378894) for the 2022 season [25,26].

### 2.3. Phylogenetic Analysis

To differentiate the genetic clades of circulated Myanmar viruses from those of viruses from other countries, genomic similarity and geographical topology were determined using an open-source database of the public health pathogen genome, Nextstrain (https://nextstrain.org, (accessed on 15 February 2023)). Nextstrain provides continually updated publicly available data enabled by GISAD. Trees were automatically built based on the main antigenic protein HA or NA, and countries or clades were automatically differentiated by colors based on the features of the website [27,28]. The time scale was set to 2 years, which was the default. Influenza vaccine strains were automatically determined by the website and marked by crosses [25,26]. Branches with Myanmar viruses were magnified to demonstrate their genetic proximity to strains from other countries, time of virus origin, and plausible roots. To detect possible reassortment events for influenza A(H3N2) Myanmar viruses, we compared the phylogenetic topology between hemagglutinin and neuraminidase by selecting the second tree option and “show tanglegram” function [27]. Similarly, a comparative phylogenetic tree with tanglegrams was automatically created by Nextstrain for the B/Victoria virus [28]. To confirm the topology of HA and NA indicated by Nextstrain and to assess reassortment in other segments, we constructed Maximum Likelihood (ML) phylogenetic trees for all eight segments for A(H3N2) and B/Victoria. The reference and vaccine strains were downloaded according to the WHO Worldwide Influenza Centre (WIC) report from the Crick Institute (London, UK) [29]. Reference accession numbers for A(H3N2) phylogenetic trees are as follows: A/Hong Kong/2671/2019 EPI ISL 412381, A/Kansas/14/2017 EPI ISL 403059, A/Switzerland/9715293/2013 EPI ISL 166310, A/Florida/02/2021 EPI ISL 7260207. Reference accession numbers for B/Victoria phylogenetic trees: B/Paris/9878/2020 EPI ISL 3535244, B/Brisbane/60/2008 EPI ISL 129017. Additional viruses were randomly selected and downloaded from the GISAID between January 2021 and October 2022 from different countries, if they showed high proximity to our strains according to Nextstrain phylogenies. The trees were constructed by free Molecular Evolutionary Genetics Analysis (MEGA) software 6.0, using the maximum likelihood method with 1000 bootstrap replications (USA, Texas) [30]. The best statistical model was determined for each segment by the lowest Akaike information criterion (AIC) value.

### 2.4. Ethical Considerations

This study was approved by the Niigata University School of Medicine Ethical Committee (No. 2015-2533) and Ethical Review Committee of the Department of Medical Research, Ministry of Health and Sports, Myanmar (No. 006EEA). Written consent was obtained from the parents or guardians of all participants before sample collection.

## 3. Results

### 3.1. Number of Samples and Detection of Influenza Virus Types and Subtypes

An increasing number of ILI cases at the end of September 2021 in Yangon was noted by local clinicians (Figure 1). During the study period, 414 patients with ILI symptoms who visited the outpatient clinic from October to December 2021 were screened by RDT. A total of 147 RDT-positive samples were collected from ILI cases in this study. The majority were positive for RS virus (n = 114, 77.6%) according to a BioFire^®^ FilmArray^®^, whereas the remaining samples were positive for influenza A(H3N2) (n = 26, 17.7%) and influenza B (n = 4, 2.7%), three specimens were negative. Among the 26 samples that were positive for influenza A(H3N2) 13 showed C*t* values ≤ 38 and were considered positive, while 10 samples showed C*t* values above the threshold (>38) and were considered negative based on the cycling probe RT-PCR assay. Of the four influenza B-positive samples, three were B/Victoria and one was negative based on a cycling probe RT-PCR assay. Influenza A(H3N2) was predominant in October and November, whereas, influenza B/Victoria was detected only at the end of November 2021 (Figure 1). The National Influenza Surveillance conducted by the National Health Laboratory in Myanmar reported similar influenza activity with our findings to the WHO Flu Net: 12 A(H3N2) positive and 1 B/Victoria case in October 2021 [3]. In our BioFire^®^ FilmArray^®^-tested samples, co-infection with SARS-CoV-2 was not detected; however, two influenza A(H3N2) and 27 RSV cases were co-infected with human rhinovirus (HRV) and one RSV patient was co-infected with adenovirus.

### 3.2. Clinical Characteristics of Patients in This Study

Herein, the clinical information of 16 patients who were positive based on the cycling probe RT-PCR assay were analyzed. The patients ranged in age from 6 months to 7 years. All of the included patients had cough, 12 had rhinorrhea, and 7 had body temperature ≥ 37.5 °C. Only one patient developed headache and diarrhea. Notably, although 1 patient received an influenza vaccine, none of the participants received antiviral therapy (Table 1).

### 3.3. Next-Generation Sequencing of Influenza Viruses Collected in Myanmar in 2021

All of the 16 cycling probe RT-PCR-positive samples, including 13 A(H3N2) samples and 3 B/Victoria samples, underwent next-generation sequencing and were mapped to the reference strains. The generated fasta files of eight segments from each strain were registered in GISAID (Appendix A).

#### 3.3.1. Genetic Analysis of Influenza A(H3N2)

The phylogeny of the HA segment, including global influenza A(H3N2) strains generated by Nextstrain, demonstrated predominant circulation of clade 2 and its subclades from August 2020 and co-circulation of clades 1, 1a and 3C.2a1b.1a. Clade 2 was descended from clade 3C.2a1b.2a, which was defined by the Y159N, T160I, L164Q, G186D, and D190N substitutions in the HA protein [31]. All 13 A(H3N2) viruses collected in Myanmar in 2021 were localized in a group, clade 2a.3 but differed from the SH 2021 corresponding season vaccine strain A/Hong Kong/45/2019, which belongs to clade 3C.2a1b.1b [25]. Similarly, the vaccine strain for the 2021–2022 Northern Hemisphere (NH) influenza season, A/Cambodia/e0826360/2020 fell into a different clade, 1a (Figure 2a) [32]. Thus, the WHO changed the recommended A(H3N2) vaccine strains for the following 2022 (SH) and 2022–2023 (NH) influenza seasons into A/Darwin/9/2021 for egg-based vaccine and an A/Darwin/6/2021-like virus, a cell-based vaccine, categorized under the clade 2a (Figure 2a) [26,33]. The Nextstrain analysis of the HA segment showed that the viruses from Myanmar were close to those from Bahrain, Australia, Qatar, Sri Lanka, Bangladesh and Nepal, which belong to Middle East and South Asia, circulated between June 2021 and January 2022 (Figure 2b).

To further investigate the genomic changes in the viruses collected in Myanmar in 2021, we compared eight segments of the influenza A(H3N2) viruses sequenced in this study with the SH 2021 vaccine component A/Hong Kong/45/2019 and the following SH 2022 cell-based vaccine component A/Darwin/6/2021 (Table 2 and Table 3, Appendix A). Comparison of the HA segment between Myanmar viruses and the SH 2021 vaccine strain demonstrated 21 common amino acid changes (Table 2) [29]. On the other hand, comparison of the HA segment with the SH 2022 vaccine strain revealed fewer differences and five common substitutions (D53N, N96S, I192F, G225D, and N378S) (Table 3). Analysis of the remaining seven segments (PB2, PB1, PA, NP, NA, MP, and NS) also revealed a high number of amino acid differences between influenza A(H3N2) virus from Myanmar and the SH 2021 vaccine strain and a low number of mismatches with the SH 2022 (Appendix A). These findings indicate that genetic mismatches were resolved in the newly selected vaccine strain.

Next, we assessed the evolution of influenza A(H3N2) Myanmar viruses between the HA and NA segments created by Nextstrain [27]. The trees demonstrated similarity in topology between the HA and NA segments for the viruses collected in 2021 in Myanmar, suggesting that a major reassortment did not occur in the A(H3N2) Myanmar viruses (Figure 3).

To confirm the topology that was found using Nextstrain, we constructed ML phylogenetic trees for each of eight segments by MEGA6 (Appendix A). Along with the SH 2022 vaccine candidate and reference A/Florida/02/2021, viruses from Myanmar fell into the same large branch, whereas the SH 2021 vaccine strain was located in a different branch among eight segments for PB2, PB1, PA, HA, NP, NA, and MP (Appendix A). However, in the phylogeny of the NS segment of Myanmar viruses, SH 2021 and SH 2022 vaccine strains, and reference A/Florida/02/2021 belonged to one branch (Appendix A).

#### 3.3.2. Genetic Analysis of Influenza B/Victoria

Globally, in the 2021 influenza season, B/Victoria viruses formed the predominant clade V1A.3a.2, with co-circulation of clades V1A.3a.1 and V1A.3a (Figure 4a). Clade V1A.3a.2 had three amino acid deletions in HA (positions 162–164) and was defined by A127T, P144L, and K203R substitutions. All Myanmar viruses belonged to clade V1A.3a.2 (Figure 4a). The WHO-recommended influenza vaccine component for the SH 2021 and NH 2021–2022 seasons for B/Victoria lineage, a B/Washington/02/2019-like virus, belonged to clade V1A.3, which was distinct from the Myanmar viruses circulating in 2021 (Figure 4a). In contrast, the WHO-recommended vaccine component of the next 2022 (SH) and 2022–23 (NH) influenza season, B/Austria/1359417/2021-like virus, belonged to clade V1A.3a.2 and matched the majority of the globally circulating viruses during 2021–2022, as well as those collected in Myanmar in 2021 (Figure 4a) [33]. We found that Myanmar strains had the highest proximity to Bangladesh strains, indicating the possible transmission route of influenza B/Victoria virus to Myanmar. Likewise, viruses from Bahrain, South Africa, Qatar, and New Zealand, as with those from Bangladesh, have genetic similarities with those collected in Myanmar, indicating that an interrelation occurred (Figure 4b,c).

After comparing the HA segments of Myanmar viruses and SH 2021 vaccine component, B/Washington/02/2019, eight amino acid mismatches were found (Table 4). Among these, three amino acid substitutions, A127T, P144L, and K203R, define clade V1A.3a.2 [29,31,34]. Additionally, N150K was detected in all the collected Myanmar viruses. The SH 2022 vaccine B/Austria/1359417/2021, unlike the SH 2021 strain B/Washington/02/2019, harbors substitutions at both 150 and 203 positions similar to the Myanmar viruses in 2021. A comparison of the HA segments of the influenza B/Victoria viruses from Myanmar with those of SH 2022 vaccine did not detect any common amino acid differences, highlighting a match with the newly selected vaccine strain.

Genetic analysis of the rest of seven segments (PB2, PB1, PA, NP, NA, MP, and NS) confirmed a high number of amino acid differences between the Myanmar viruses and the SH 2021 B/Washington/02/2019 vaccine strain, and fewer mismatches with the SH 2022 B/Austria/1359417/2021 (Appendix A). These findings further confirmed the similarity between the B/Victoria vaccine strain and viruses collected in Myanmar in 2021.

To investigate the evolution of B/Victoria viruses, we compared the tanglegram of the HA and NA segments of the viruses from Myanmar and vaccine strains of the 2021 and 2022 seasons by Nextstrain [28]. We found congruent topology between the HA and NA segments for Myanmar viruses and vaccine strains, which fell into one the largest circulating groups in clade V1A.3a2 (Figure 5). ML phylogenetic trees, constructed for each of eight segments by MEGA6, also revealed that Myanmar viruses grouped in clade V1A.3a.2 with SH 2022 vaccine strain, a reference strain of B/Paris/9878/2020 and other country strains and no reassortment event was detected (Appendix A).

## 4. Discussion

Herein, we reported atypical influenza activity caused by influenza A(H3N2) and B/Victoria in Myanmar between October and November 2021 during the COVID-19 pandemic. Circulated viruses were antigenically distinguished from vaccine strains, but the mismatch was resolved in the following season owing to changes in the vaccine strains. The circulating viruses were genetically similar to strains circulating in the Middle East and South Asia during the same period. Our data suggest the possible interrelation of influenza A(H3N2) viruses between Myanmar and the Middle East and the interrelation of B/Victoria viruses within South Asia. This is the first study to report the molecular-epidemiology of influenza virus based on whole-genome analyses in Myanmar during the COVID-19 pandemic.

In this study, we observed irregular influenza activity in 2021 in the Yangon region of Myanmar, which was detected in October and ended in November. Before the COVID-19 pandemic, Myanmar experienced influenza outbreaks during the rainy season peaking in July [3,7]. Since 2020, the COVID-19 infection has become the main threat to the global population, and the public health measures to suppress the spread of COVID-19 have resulted in decreased influenza activity in Asian countries [4,35,36,37]. Bangladesh and China, which are neighboring countries of Myanmar, experienced the phenomenon of limited influenza circulations in 2020, suggesting the significant effect of restrictions controlling COVID-19 and reducing seasonal influenza transmission [38,39]. However, influenza activity gradually revived in some Asia-Pacific countries in 2021 after public health measures were lifted and based on the reduced immunity against influenza viruses after the low epidemic period of influenza for over a year during the COVID-19 pandemic [3,34,40]. Indeed, an influenza virus epidemic did not occur in Myanmar during 2020 and up until the middle of 2021, although a sudden rise of influenza and RSV cases was observed in late 2021 after the community outbreak of the delta variant of SARS-CoV-2 began to decline. According to the WHO surveillance database, in 2021, Australia, presented the co-circulation of A(H3N2) and B/Victoria viruses [3]. Qatar experienced influenza B/Victoria circulation in late February 2021, followed by the prominent circulation of A(H3N2) and a few A(H1N1)pdm09 viruses [3]. A similar pattern of influenza circulation was observed in Bangladesh, where, after the circulation of B/Victoria from April to August, it was replaced by the predominant circulation of A(H1N1)pdm09 and few A(H3N2) viruses [3]. In Nepal, however, an influenza outbreak started in July with dominant circulation of influenza A(H3N2) circulation and few A(H1N1)pdm09 viruses, gradually changing to the influenza B in 2021 [3]. Only a few influenza A(H3N2) strains have been detected in Sri Lanka in 2021 [3].

Our previous study focused on phylogeographic analyses of the influenza virus in 2010–2015 and suggested that the origins of Myanmar influenza A(H3N2) virus could be Europe with a core transmission hub of Australia [7]. In this study, we speculate that Myanmar influenza A(H3N2) virus in 2021 can be transmitted from the Middle East and surrounding South Asian countries. Additionally, our previous study showed a strong path of B/Victoria from Singapore to Myanmar [7]. However, our present findings display that B/Victoria virus was transmitted to Myanmar through Bangladesh, with alternative possible routes being Singapore and Bahrain. Based on these findings, we hypothesized a source of transmission for the influenza virus was altered after the COVID-19 pandemic based on the restrictions on international travel and implementation of personal hygiene measurements, such as mask wearing. For the current study, we used a ready-made dataset in Nextstrain; however, we further intend to analyze the changes in transmission routes of influenza viruses to Myanmar in the post-pandemic era by constructing our own dataset to run using the Nextstrain algorithm.

Our genetic and phylogenetic analyses confirmed mismatches of the influenza A(H3N2) SH 2021 vaccine with the Myanmar strains and majority of viruses circulating globally in 2021. The HA amino acid substitutions found in this study are located in the antigenic and receptor binding sites (131 is close to the receptor-binding site, 135 is located on antigenic site A, 159 and 160 in antigenic site B), resulting in the gain of a glycosylation site that is poorly recognized by ferret antisera due to antigenic changes [26,39,41,42]. Nevertheless, an updated vaccine strain, in concordance with circulating viruses recognized by ferret antisera, was selected for the SH 2022 influenza season [26].

Influenza B/Victoria viruses from our study possessed three amino acid deletions, as observed in recent global strains; they belong to the main clade V1A.3a2. A 3-D structural model indicates that all substitutions lay atop the globular head of the HA and flank the HA receptor-binding site (RSB) [43]. The K203R substitution has an impact on the antigenic properties of the virus, and the combination of N150K and K203R in HA demonstrates a similar effect [43]. Our viruses were consistent with the vaccine strain selected for the following 2022 season, and mismatches were resolved [26,29].

Significant genetic diversity of the B/Victoria lineage was observed between the HA and NA segments in circulating viruses in 2021–2022. Thus, B/Victoria viruses from China that circulated in 2021–2022 formed two genetic groups and fell into different clades. One formed a group by developing an additional H122Q substitution and underwent interlineage reassortment [29,34]. Despite being from the same genetic clade, Myanmar viruses that belonged to different groups evolved without the assistance of reassortment [44]. Another large group of viruses from China belonged to a different clade and did not exhibit reassortment events [29,31,34]. A previous study with a large dataset from 1986 to 2006 showed that the genesis of most new HA clades of influenza B is associated with reassortment, which can involve parental viruses from different antigenic lineages or a single lineage and reassortment events involve a new combination of HA and NA segments, with potentially important effects on viral activity [45]. Taken together, these findings emphasize the importance of global monitoring of influenza viruses, including whole-genome analysis.

Before the COVID-19 pandemic, the major patterns of influenza circulation in tropical countries in Asia presented influenza peak activity between June and October; therefore, the best time for vaccination was in May. However, the current schedule in most tropical countries in Asia is between June and December, which was found to be suboptimal [46,47]. Influenza vaccinations should precede peak activity periods to confer maximum protection; therefore, tropical and subtropical countries of southern and southeastern Asia should consider starting vaccination campaigns earlier relative to other temperate climate countries in the Northern Hemisphere. Such campaigns will require national policy makers to observe influenza circulation patterns each year and use the most recent WHO-recommended vaccine formulation provided immediately before the start of the influenza season [14,46,47]. However, we should continue to monitor the changes in the influenza circulating season in each country because it was greatly affected by the COVID-19 pandemic.

This study has several limitations. First, a risk of oversampling exists since the data uploaded data in GISAID were used to construct the phylogenetic trees and some countries have submitted more sequences in the database, while other countries have fewer sequences. As a result, our conclusions could be biased. Second, the small sample size in this study, which led to a small number of successfully sequenced samples did not allow to demonstrate comprehensive genetic variations. Third, to provide a more detailed and focused analysis, we did not generate our own genetic dataset for influenza viruses to run in the Nextstrain interface, and we did not use an individual software package, such as BEAST, for the phylogenetic analysis with an emphasis on time-scaled trees or for geo-temporal map construction with estimated transmission routes [48]. Fourth, the influenza A(H3N2) samples in this study demonstrated that the cycling probe RT-PCR presented low sensitivity compared with the BioFire^®^ FilmArray^®^ multiplex RT-PCR (26/13), which was likely because of the low viral loads (1.5–2.9 log_10_) of the cycling probe-negative samples relative to the high viral loads (3.0–4.7 log_10_) in the cycling probe-positive samples, as confirmed by high-sensitivity quantitative RT-PCR using TaqMan probes [49].

## 5. Conclusions

This is the first study to report epidemiological and genetic analyses of influenza viruses that occurred in Myanmar in 2021 during the COVID-19 pandemic. This highlights the importance of continuing global influenza surveillance. Viruses collected in Myanmar can contribute to the detection of the new genetic changes and selection of global influenza vaccines, understanding transmission, and control and prevention of influenza. We believe that the present findings will be beneficial to the public health sectors.

## Figures and Tables

**Figure 1 viruses-15-00583-f001:**
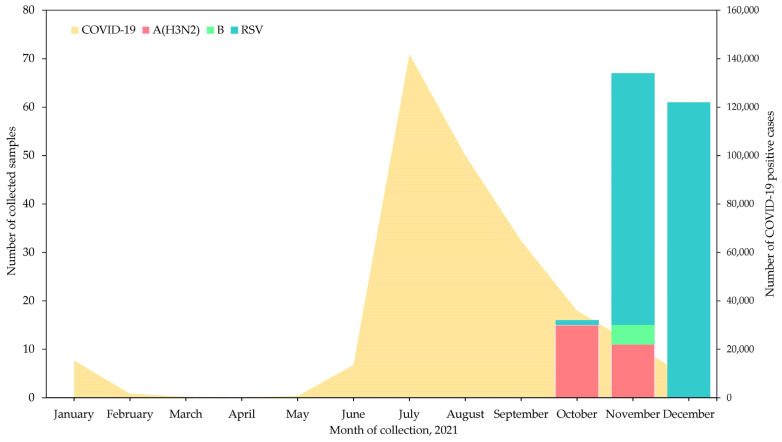
Epidemic curve of influenza A(H3N2), B and respiratory syncytial virus (RSV) confirmed with BioFire^®^ FilmArray^®^ RT-PCR in this study and the official number of COVID-19 cases from the Myanmar COVID-19 surveillance dashboard of the Myanmar Ministry of Health in Yangon, 2021 (https://mohs.gov.mm/Main/content/new/list?pagenumber=1&pagesize=9, (accessed on 16 December 2022)).

**Figure 2 viruses-15-00583-f002:**
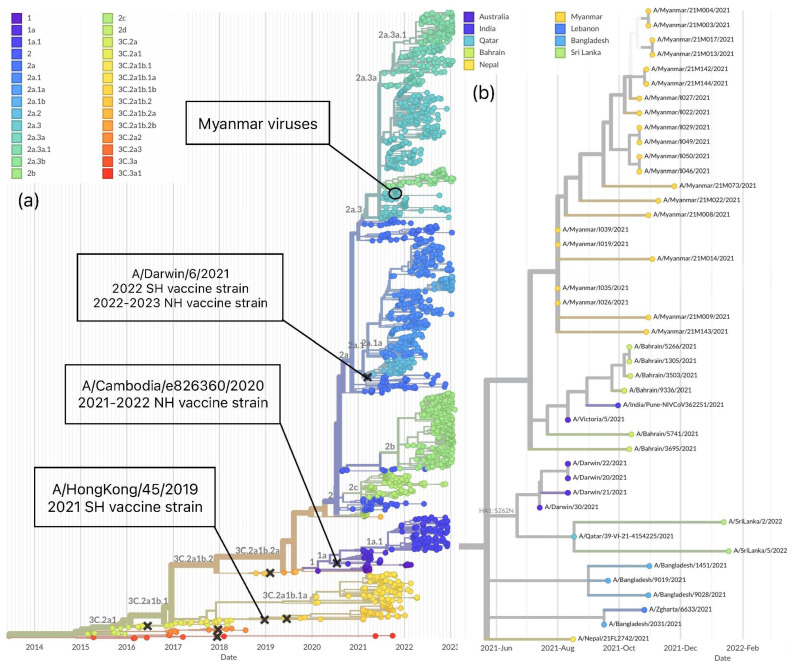
Time-aware phylogeny of the HA segment of the influenza A(H3N2) viruses constructed by Nextstrain (accessed on 17 February 2023) [27]. (**a**) Clade differentiation of the globally circulated influenza A(H3N2) based on the HA segment. The vast majority of circulated viruses worldwide and Myanmar viruses in 2021, are categorized under clade 2, circulation of three different clades were observed: 3C.2a1b.1a, 1 and 1a. Vaccine strains for SH 2021 belonged to clade 3C.2a1b.1b and those for NH 2021–2022 to the clade 1a. Newly selected vaccine candidate for SH 2022 and NH 2022–2023 seasons belongs to the clade 2a. (**b**) Close look at the Nextstrain phylogeny including influenza A(H3N2) viruses in Myanmar and genetically similar viruses in other countries during 2021 and 2022. Strains, collected in Myanmar in 2021 are similar to those found in Bahrain, Australia, Qatar, Sri Lanka, Bangladesh, and Nepal.

**Figure 3 viruses-15-00583-f003:**
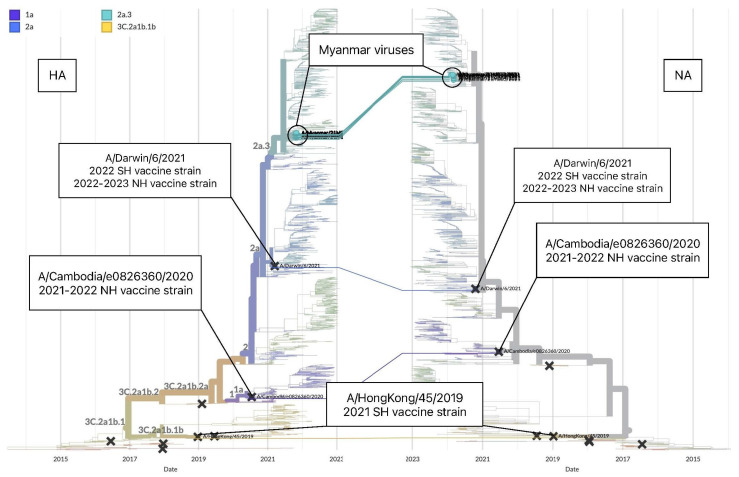
Tanglegram of the HA and NA segments of the influenza A(H3N2) viruses with a time scale of 2 years displayed by Nextstrain (accessed on 17 February 2023) [27]. Comparative topology between the HA and NA segments for influenza A(H3N2) viruses collected in Myanmar during this study and vaccine strains of the 2021 and 2022 seasons.

**Figure 4 viruses-15-00583-f004:**
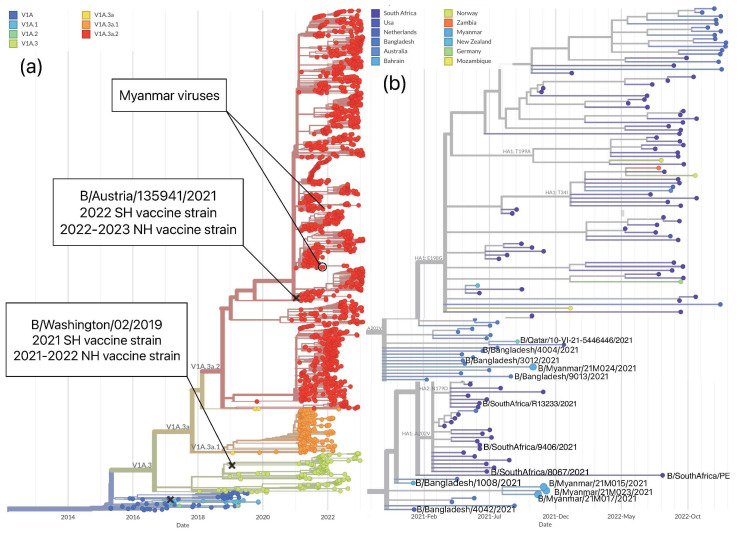
Time-aware phylogeny of the HA segment of the influenza B/Victoria viruses in Myanmar constructed by Nextstrain (accessed on 17 February 2023) [28]. (**a**) Clade differentiation of the HA segment of the B/Victoria viruses. The majority of circulated viruses collected in 2021 fell into clade V1A.3a.2 with co-circulation of clades V1A.3a.1 and V1A.3. All Myanmar viruses belong to clade V1A.3a.2. The same vaccine strain B/Washington/02/2019 belonging to clade V1A.3 was selected for the SH 2021 and NH 2021–2022 influenza seasons. The 2022 SH and 2022-23 NH vaccine strain B/Australia/135941/2021 fell into clade V1A.3a.2. (**b**) Close look at the Nextstrain phylogeny including Myanmar B/Victoria viruses and genetically similar viruses in other countries during 2021 and 2022. Strains collected in Myanmar 2021 are similar to those in Bangladesh, Bahrain, South Africa, and Qatar.

**Figure 5 viruses-15-00583-f005:**
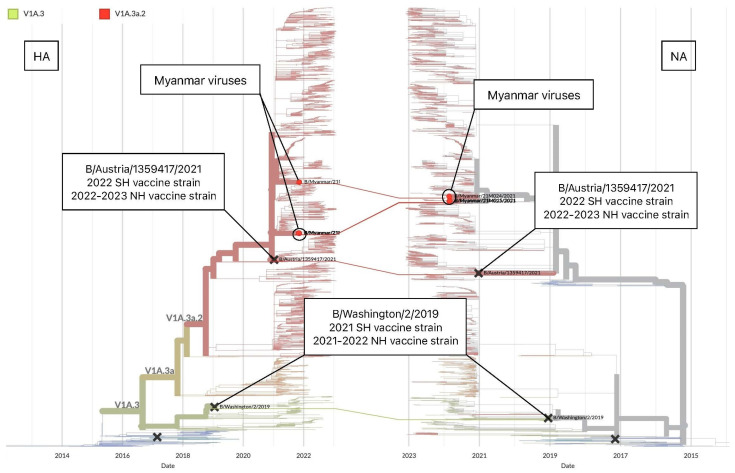
Tanglegram of the HA and NA segments of B/Victoria viruses collected in November 2021 in Myanmar with a time scale of 2 years displayed by Nextstrain (accessed on 17 February 2023) [28]. Comparative topology of the HA and NA segments of the B/Victoria viruses collected in 2021 in Myanmar during this study and vaccine strains of the 2021 and 2022 seasons.

**Table 1 viruses-15-00583-t001:** Baseline and clinical characteristics of A(H3N2)- and B/Victoria-infected patients confirmed by cycling probe RT-PCR, Myanmar, 2021.

Type (Subtype/Lineage)	A(H3N2) (n = 13)	B/Victoria (n = 3)
Variables		
Age, median [min–max] (years)	2 [0.5–7]	2 [1–4]
Gender, male (%)	7 (53.8)	3 (100)
Symptoms, presence (%)		
Cough	13 (100)	3 (100)
Fever ≥ 37.5 °C	5 (38.5)	2 (66.6)
Rhinorrhea	10 (76.9)	2 (66.6)
Headache	0 (0.0)	1 (33.3)
Diarrhea	0 (0.0)	1 (33.3)
Influenza vaccination		
Presence (%)	1 (7.7)	0 (0.0)

**Table 2 viruses-15-00583-t002:** Common amino acid substitutions found in the HA segment of A(H3N2) viruses collected in Myanmar in 2021 compared with those of the SH 2021 vaccine strain A/Hong Kong/45/2019.

		Amino Acid Substitutions										
	Segment	HA										
Vaccine strain	Clade	53	83	94	96	128	131	135	137	138	156	159 *
A/Hong Kong/45/2019	3C.2a1b.1b	D	K	Y	N	A	T	K	F	S	H	Y
Myanmar viruses 2021	2a.3	N	E	N	S	T	K	T	S	A	S	N
	**Segment**	**HA**										
Vaccine strain	Clade	160 *	164	186	190	192	195	312	378	522	529	
A/Hong Kong/45/2019	3C.2a1b.1b	T	L	G	D	I	Y	N	N	I	V	
Myanmar viruses 2021	2a.3	I	Q	D	N	F	F	S	S	M	I	

Abbreviations: HA, hemagglutinin; SH, Southern Hemisphere. * Indicates the substitutions with loss of the glycosylation sites.

**Table 3 viruses-15-00583-t003:** Common amino acid substitutions in the HA segment of A(H3N2) viruses collected in Myanmar in 2021 compared with those of the SH 2022 vaccine strain A/Darwin/6/2021.

		Amino Acid Substitutions				
	Segment	HA				
Vaccine strain	Clade	53	96	192	225	378
A/Darwin/6/2021	2a	D	N	I	G	N
Myanmar viruses 2021	2a.3	N	S	F	D	S

Abbreviations: HA, hemagglutinin; SH, Southern Hemisphere.

**Table 4 viruses-15-00583-t004:** Common amino acid substitutions in the HA segment of the B/Victoria viruses collected in Myanmar in 2021 compared with those of the SH 2021 vaccine strain B/Washington/02/2019.

		Amino Acid Substitutions							
	Segment	HA							
Vaccine strain	Clade	127	133	144	150	181	194	203	276
B/Washington/02/2019	V1A.3	A	R	P	N	G	N	K	R
Myanmar viruses 2021	V1A.3a.2	T	G	L	K	E	E/D	R	K

Abbreviations: HA, hemagglutinin; SH, Southern Hemisphere.

## Data Availability

The whole-genome sequences generated in this study have been deposited in the Global Initiative on Sharing All Influenza Data (GISAID) EpiFlu database and the accession numbers are listed in Appendix A.

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
