# Peer review of "Whole-Genome Analysis of Influenza A(H3N2) and B/Victoria Viruses Detected in Myanmar during the COVID-19 Pandemic in 2021"

_viruses, 2023, doi:10.3390/v15020583_

Round 1

Reviewer 1 Report

Chon et al. performed the surveillance of influenza viruses in Myanmar under the backgroundCOVID-19 pandemic in 2021. I think that this study has potential implications for understanding the evolutionary feature of influenza viruses and the vaccine update. However, this manuscript still has many issues that need to be solved by the authors especially for their poor figure plotting, which are hard to publish. In all sincerity, their scientific theme is useful and important for current area.

I mainly concern the following some issues:

1. The authors should provide a clear background information about 147 samples you collected. For example, how many patients were screened to obtain 147 samples with symptoms of influenza-like illness? I want to estimate your screening base size because a small population may result in biases. Otherwise, I can not evaluate the level of whole influenza circulation in Myanmar during the the COVID-19 pandemic in 2021.

2. In this study, did the authors find the cases with the dual infection by influenza viruses and SARS-CoV-2? If so, this may be a highlight of the article.

3. In 3.1 of Results, the authors have finished the detection of influenza virus types and sub-types and found 16 samples as influenza A and 3 samples as B/Victoria by cycling probe RT-PCR. Therefore, for 3.2 of Results, the authors should use the 16+3 information for the downstream statistics of clinical characteristics according to the progressive logical relationship because other samples belonged to false positives.

4. Figs 1-5 in the main text should be difficult to publish because of extremely low resolution. I suspect authors directly obtained a screenshot from the Nexstrain web. I think that this situation should be avoided and this involves in many issues. The authors should provide vector diagrams and write clearly methods for the constructed tree (e.g., bootstraps).

5. Can you give an explanation why did you choose the SH vaccines for comparison (not NH) when you analyzed influenza A? It is confusing especially for the peers focusing on animal genomics and evolution.

6. Your fig 4b-4c are very unclear with the severe legend occlusion.

7. Authors should do further functional analyses or predictive parsing for those amino acid mutations to make your analyses more rich.

8. The authors should be cautious when describing the direction of virus transmission in lines 374-375.

Some minor issues also should be revised well when they submit their revised manuscript.

1. I suggest that the authors remove the company or reagent information in abstract (line 28), e.g., ‘Bio-Fire® FilmArray®’, because the abstract should highlight the most important results or conclusions.

2. The authors did not uniform the font size in lines 202-203.

3. In line 231, the authors should use ‘3C.2a1b.1b’ instead of ‘3C.2a.1b.1b’. I think the clade name should be consistent through the manuscript.

4. I suggest authors remove the - in line 232.

5. In line 232, the authors should use 3C.2a1b.2a.1 instead of 3C.2a.1b.2a.1. The clade name should be consistent through the manuscript.

6. Authors should bring the clade name of vaccine strain in table 2 into correspondence with the Figure 2 to avoid confusion.

7. I suggest the authors remove the word ‘glycoproteins’ in line 294 because of good-for-nothing.

8. I suggest the authors remove the - in line 328 because of futility.

9. I suggest that authors remove the table 5 because of futility.

Reviewer 2 Report

The article about " Whole genome analysis of influenza A(H3N2) and B/Victoria viruses detected in Myanmar during the COVID-19 pandemic in 202" was showed as an important exploratory study about the main subjects of a intersting co-infection influenza virus and SARSCoV2. The authors could exploiting mores those interactions in discussion. Also, the data concerning the NGS run in this article could better explicated the high diversity of viurus detected in this study. Some explanation about this result? 

Congratulations for a technically great effort for those work.

Author Response

Dear Reviewer,

thank you very much for kind and encouraging feedback.

We revised the manuscript and improved description and discussion part. We hope that the revised manuscript is highlighting all questions of interest.

Sincerely,

Chon Irina

Division of International Health (Public Health)

Graduate School of Medical and Dental Sciences, Niigata University

1-757 Asahimachi-dori, Chuo-ward, Niigata City, Niigata Prefecture, 951-8510 Japan.

Phone: +81-25-227-2129

Fax: +81-25-227-0765.

E-mail: irinachon@med.niigata-u.ac.jp

Reviewer 3 Report

This study investigated the epidemiological and genetic analyses of influenza viruses that occurred in Myanmar in 2021 during the COVID-19 pandemic.

The manuscript well written besides excellent bioinformatic analysis. Further studies like this needs to be conducting to deeply investigate the correlation between COVID-19 and other respiratory viruses in different countries besides updating used vaccinal strains for prevention.

Line 37: genetic characterization of the influenza virus in Myanmar to contribute to global influenza surveil…

I prefer to modify it : genetic characterization of the influenza virus in Myanmar  for  contribution in to global influenza surveil…

Line 46: Toward the end of 2019, severe acute respiratory 46

I prefer to modify it : At the end of 2019, severe acute respiratory syndrome coronavirus 2 (SARS-CoV-2..

Line 196-97 : Among 26 samples found positive for influenza A(H3N2) using BioFire® FilmArray® , were confirmed to be positive by cycling probe RT-PCR assay, and 10 samples were negative

What is the author explanation ! and what about the test sensitivity? Also what about the sensitivity of used primers in RT-PCR?

Fig 1 there are reverse relation between COVID-19 and other respiratory viruses?

What about the author Explanation? I think the authors should discuss that and possible explanation in part of discussion

Line 217: Of the 29 RT-PCR-positive samples, 16 samples with Ct values < 38, found by cycling

For NGS without virus isolation samples ct must be lower than 30 ct

I think most of figures needs better resolution

Round 2

Reviewer 1 Report

The authors have settled my concerns. I recommend the manuscript for publication.